# Quantitative Analysis of Isopimpinellin from *Ammi majus* L. Fruits and Evaluation of Its Biological Effect on Selected Human Tumor Cells

**DOI:** 10.3390/molecules29122874

**Published:** 2024-06-17

**Authors:** Magdalena Bartnik, Adrianna Sławińska-Brych, Magdalena Mizerska-Kowalska, Anna Karolina Kania, Barbara Zdzisińska

**Affiliations:** 1Department of Pharmacognosy with Medicinal Plants Garden, Medical University of Lublin, Chodźki 1 Street, 20-093 Lublin, Poland; amazurek02@interia.pl; 2Department of Cell Biology, Institute of Biological Sciences, Maria Curie-Skłodowska University, Akademicka 19 Street, 20-033 Lublin, Poland; adrianna.slawinska-brych@mail.umcs.pl; 3Department of Virology and Immunology, Institute of Biological Sciences, Maria Curie-Skłodowska University, Akademicka 19 Street, 20-033 Lublin, Poland; magdalena.mizerska-kowalska@mail.umcs.pl (M.M.-K.); barbara.zdzisinska@mail.umcs.pl (B.Z.)

**Keywords:** isopimpinellin, *Ammi majus* fruits, ASE extraction, HPLC/DAD quantitative analysis, LC/CPC isolation, antiproliferative activity, apoptosis, osteosarcoma (Saos-2 and HOS), multiple myeloma (RPMI8226 and U266), colorectal adenocarcinoma (HT29 and SW620)

## Abstract

*Ammi majus* L. (Apiaceae) is a medicinal plant with a well-documented history in phytotherapy. The aim of the present work was to isolate isopimpinellin (5,8-methoxypsoralen; IsoP) from the fruit of this plant and evaluate its biological activity against selected tumor cell lines. The methanol extract obtained with the use of an accelerated solvent extraction (ASE) method was the most suitable for the quantitative analysis of coumarins in the *A. majus* fruit matrix. The coumarin content was estimated by RP-HPLC/DAD, and the amount of IsoP was found to be 404.14 mg/100 g dry wt., constituting 24.56% of the total coumarin fraction (1.65 g/100 g). This, along with the presence of xanthotoxin (368.04 mg/100 g, 22.36%) and bergapten (253.05 mg/100 g, 15.38%), confirmed *A. majus* fruits as an excellent source of these compounds. IsoP was isolated (99.8% purity) by combined liquid chromatography/centrifugal partition chromatography (LC/CPC) and tested for the first time on its antiproliferative activity against human colorectal adenocarcinoma (HT29, SW620), osteosarcoma (Saos-2, HOS), and multiple myeloma (RPMI8226, U266) cell lines. MTT assay results (96 h incubation) demonstrated a dose- and cell line-dependent decrease in cell proliferation/viability, with the strongest effect of IsoP against the Saos-2 cell line (IC50; 42.59 µM), medium effect against U266, HT-29, and RPMI8226 (IC50 = 84.14, 95.53, and 105.0 µM, respectively), and very weak activity against invasive HOS (IC50; 321.6 µM) and SW620 (IC50; 711.30 µM) cells, as well as normal human skin fibroblasts (HSFs), with IC50; 410.7 µM. The mechanistic study on the Saos-2 cell line showed that IsoP was able to reduce DNA synthesis and trigger apoptosis via caspase-3 activation. In general, IsoP was found to have more potency towards cancerous cells (except for HOS and SW620) than against healthy cells. The Selective Index (SI) was determined, underlining the higher selectivity of IsoP towards cancer cells compared to healthy cells (SI = 9.62 against Saos-2). All these results suggest that IsoP might be a promising molecule in the chemo-prevention and treatment of primary osteosarcoma.

## 1. Introduction

Nowadays, a shift towards natural medicine can be observed, as can the exploration of the potential of plant-derived compounds as active agents in the treatment of various ailments, including cancer, as an alternative to synthetic drugs [1,2,3]. As a natural consequence, we observed continuously increasing interest in the isolation and purification of compounds of plant origin, and their further evaluation as chemo-preventive and anticancer agents can be observed. Especially ecologically friendly methods, such as pressurized solvent extraction (PLE) also called accelerated solvent extraction (ASE), are of great importance [4,5,6]. The lower time and solvent consumption, compared to classical extraction methods (e.g., the still frequently used tedious Soxhlet extraction) and multistep chromatographic methods of isolation, make ASE [7] and centrifugal partition chromatography (CPC) combined together an attractive way for the isolation of bioactive molecules from plant matrices [8]. Additionally, the easy transition from the analytical to the preparative scale is a big advantage of the CPC methodology [9,10,11,12]. CPC has been successfully used for the isolation of coumarins many times before, including in the work of our team [12,13].

The species *Ammi majus* L. (Apiaceae) is a medicinal plant originating from Egypt and is widely distributed in Europe, the Mediterranean, and Western Asia. It has a well-documented history in ethnomedicine [14,15]. Various parts of this plant (mainly the fruits, but also the leaves and whole aerial parts) have been used in the treatment of various dermatological diseases, especially vitiligo, and in traditional medicine (mainly fruits) as an emmenagogic, diuretic, and blood purifier agent in the treatment of leprosy and urinary and digestive system disorders [14]. Several review articles have recently summarized the various pharmacological effects of this important natural medicine [14,15,16,17]. This plant material serves as an excellent source of furanocoumarins, especially bergapten (5 MOP), xanthotoxin (8 MOP), and isopimpinellin (IsoP) [14,15], whose content was qualitatively and quantitatively evaluated in several studies. HPLC, LC/MS, and UPLC/MS methods were successfully developed for this purpose [13,18,19]. However, the PLE/ASE/HPLC methodology for quantitative or preparative purposes has not been used for *A. majus* fruit matrix to date, which was confirmed by the search in multiple databases (SciFinder CAS, Science Direct, Web of Science Clarivate, Medline, Pub-Med) (years 1990–2024; search topics; “(Ammi majus) and ASE”; ‘(Ammi majus) and PLE”). There is only one report by Królicka et al. on the use of the ASE method for the evaluation of the components in transformed *Ammi majus* callus cultures [20], where the content of two coumarins, umbelliferon and bergapten, was estimated with the ASE/HPLC method. 

Isopimpinellin (5,8-methoxypsoralen; IsoP, Figure 1), one of the coumarins found in *A. majus* fruits, is a bioactive molecule that exerts various documented biological activities [15]. It was proven to be active against bacteria and fungi [21,22] and acts as an effective adjuvant in antibacterial and anticonvulsant treatment [23]. Isopimpinellin is known to be a bitter taste receptor TAS2R14 agonist [24], a potent hemostatic agent [25], and a modulator of melanin production, which increases melanogenesis in mouse melanoma B16F10 cells [26]. IsoP acts as an acetylcholine esterase (AChE) inhibitor and an amyloid β (Aβ) aggregation inhibitor (AChE- and self-induced), inhibiting the AChE enzyme with IC50 = 41 µM and AChE-induced and self-induced Aβ aggregation with IC50 > 500 and 125 µM, respectively [27], which makes this compound interesting in the context of Alzheimer’s disease therapy.

Previously, Joshi et al. [28] confirmed the effect of isoP on the inhibition of cytochrome P450 enzymes, especially inhibiting the properties of the CYP1A1 isoenzyme, and the protective effect on HEK293 cells from BαP-induced toxicity (by blocking the metabolism of procarcinogens to active carcinogenic metabolites). The anti-inflammatory activity of IsoP was detected in a zebrafish in vivo model, and inhibition of phosphatidylinositol 3-kinase (PI3K) signaling by this compound (resulting in a reduction in the migration of neutrophils towards the site of injury) was observed [29]. 

However, information on the anti-proliferative and cytotoxic effects of this furanocoumarin is still very limited. Taking into account the interesting chemo-preventive properties of similar compounds from the methoxyfuranocoumarin (MFC) group (non-UV activated 8 MOP and 5 MOP) [30,31], in the present study, we decided to evaluate whether non-UVA-activated isopimpinellin may be active against tumor cell lines such as the human colorectal adenocarcinoma (HT29, SW620), osteosarcoma (Saos-2, HOS), and multiple myeloma (RPMI8226, U266), i.e., the cell lines that represent therapeutically difficult groups of malignancies with resistance to conventional therapies.

This is the first report on the estimation of the furanocoumarin content in the *A. majus* fruit matrix using the ASE/HPLC methodology and the first report on assessing the effect of isopimpinellin on the cytotoxicity, proliferation, and apoptosis induction in the selected cancer cell lines, providing yet unreported new data on the potential role of this MFC in inhibiting the growth of selected cancer cell lines.

## 2. Results

### 2.1. ASE/PLE Extraction Efficiency and Identification of Coumarins in the ASE Extract 

ASE extraction for the isolation of coumarins from the *A. majus* fruit matrix was performed. For this purpose, two solvents (dichloromethane and methanol) were tested, and ASE extracts were obtained with the use of increasing temperature values (50, 70, 90, 110, and 130). As a result, the extracts ASE/Dex/50-130 and ASE/Mex/50-130 were collected, concentrated, and carefully weighed, and the extraction yield (%) was calculated in each case (Table 1). The results indicated that the extraction yield increased with the increasing temperature in the case of each of the solvents tested. The biggest extraction yield was noted in the case of the methanol extracts at a temperature of 130 °C. The dichloromethane extracts were less efficient. The ASE methodology contributed to the use of a lower solvent volume than the previously applied Soxhlet extraction [12] and was less time-consuming. Each time, Soxhlet extraction was performed using 100 g of the plant material and lasted for c.a. 5 days after preliminary 24-hour maceration in the solvent used. The ASE extraction duration was 3 × 10 min for one ASE cycle, and with the additional operation process required (procedures of the extraction: cell loading, setting of the temperature, and cleaning of the system), each single process did not exceed 2 h.

### 2.2. Quantitative Analysis of Coumarins in PLE/ASE Extracts

For the quantitative estimation of coumarins in the *A. majus* fruit matrix with the SPE/HPLC/DAD method (the validation data are summarized in Table 2), the ASE/Mex/130 extract was chosen, and the results are presented in Table 3.

The HPLC chromatogram of the ASE/Mex/130 extract and the DAD spectra collected online during the HPLC analysis are shown in Figure 2 and Figure 3, respectively.

As a result of the quantitative analysis of coumarins in the ASE/Mex/130 extract, it was found that IsoP constituted 24.56% of the total coumarin fraction, followed by 8 MOP (22.36%) and 5 MOP (15.38%). These three methoxyfuranocoumarins were the main compounds in the investigated extract (404.14; 368.03, and 253.05 mg/100 g DW, respectively), and IMP constituted only 2.52% (41.60 mg/100 g DW); see Figure 4.

The analysis of the three main coumarins in the obtained extracts (Dex and Mex) proved that with the increasing temperature of the ASE extraction, the content of IsoP, 8 MOP, and 5 MOP increased, and methanol extracts were most effective for isolation (Table 4). 

As the HPLC analysis of the methanol extract purified by SPE showed the presence of non-coumarin polar compounds, which were in the case of the preparative process ballasts, the most convenient for the preparative isolation of targeted IsoP was dichloromethane extract (ASE/Dex/130 contained 346.53 mg/100 g DW of IsoP). This extract was then selected for preparative isolation by LC/CPC methodology [12].

### 2.3. LC/CPC Isolation and Identification of Isopimpinellin

As a result of the LC/CPC preparative steps (isolation of the SCS, semicrystalline coumarin sediment, through ASE/Dex/130 extract collection, condensation. and cooling in the fridge for 5 days, followed by filtration; the LC step to obtain fraction LC/FR6, containing the targeted furanocoumarin, followed by the CPC semi-preparative process, as described previously [12], pure isopimpinellin (99.8%) was isolated from *A. majus* fruits, and its identity and purity were confirmed by RP-HPLC/DAD and ESI-TOF MS analyses. From 30 g of the fruits, 2.12 g of the SCS was obtained, and after the LC process, 10.26 mg of the IsoP was collected from 40 mg of the LC/FR6 fraction after the CPC isolation step.

Isopimpinellin (5,8-dimethoxypsoralen; CAS 482-27-9; C_13_H_10_O_5_/246.215 Da; IUPAC name: 4,9-Dimethoxy-7H-furo[3,2-g]chromen-7-one).

HPLC/DAD: t_R_ = 14.419; DAD/UV spectra; λ_max_ = 201, 222, 250, 270, 315 nm; λ_min_ = 205, 235, 255, 278; (Figure 2 and Figure 3).

Figure 5A presents the total ion chromatogram (TIC) and the extracted ion chromatogram (EIC). The MS spectrum of the isolated isopimpinellin is shown in Figure 5B.

ESI-MS [H+] data; t_R_ = 8.571; m/z 247.0601 [M + H]^+^, 232 [M+H^+^-CH_3_], 217 [M+H^+^ -2 CH_3_]. These MS data were also confirmed by the comparison with the IsoP standard and with the literature [32,33]. Finally, the isolated isopimpinellin was used in the biological experiments described in this paper.

### 2.4. Cytostatic and Cytotoxic Activity of IsoP

The impact of isopimpinellin on the proliferation of various cancer cell lines (Saos-2, HOS, RPMI8226, U266, HT-29, and SW620) was assessed by the MTT assay. This assay is based on the metabolic conversion of the tetrazolium dye MTT, which generally correlates with the number of viable and proliferative cells. In this study, the cancer cells were exposed to either the culture medium (control) or isopimpinellin (3.125–200 µM) for 96 h. Untreated cells (control) were considered the baseline (100%). As depicted in Figure 6A–F, IsoP disrupted the mitochondrial metabolism of the examined cancer cells to a varying degree in a dose-dependent manner. This compound appeared to be the most effective against Saos-2 cells (Figure 6A) and the least active against HOS (Figure 6B) and SW620 cells (Figure 6F). The other analyzed cell lines, RPMI8226, U266, and HT-29 (Figure 6C–F), were characterized by medium sensitivity. In the case of the Saos-2 cell line, a statistically significant reduction in cell proliferation/viability was already observed at low doses starting at 3.125 µM, while a statistically significant effect was obtained in both multiple myeloma cultures (RPMI8226, U266) in the slightly higher concentration range (12.5–200 µM). In contrast, in colon cancer cells (HT-29 and SW620) and HOS cells, only the high doses (50–200 µM and 100–200 µM, respectively) produced a statistically relevant suppressive effect.

The most common indicator of the anticancer activity of an experimental compound is the half-maximum inhibitory concentration (IC50)—the dose that inhibits cell growth/viability by 50%. To demonstrate the potency of isopimpinellin, the IC50 value for each of the tumor cell lines was determined. The lower the IC_50_ values, the higher the anticancer activity, and the higher the IC50 values, the lower the anticancer activity. As estimated by the non-linear regression analyses (Figure 7), the obtained IC50 values varied due to the different responses of the cancer cell lines to the IsoP treatment. For Saos-2, it was revealed that a concentration of 42.59 µM (which corresponds to 9.84 µg/mL) induced 50% lethality, whereas in U266, HT-29, and RPMI8226 cells, this effect was provoked by much higher doses of 84.14 µM (19.69 µg/mL), 95.53 µM (23.5 µg/mL), and 105.0 µM (25.83 µg/mL), respectively. In the other two lines, HOS and SW620, the IC50 parameter reached values of 321.26 µM (79.71 µg/mL) and 711.3 µM (174.9 µg/mL), respectively. According to the National Cancer Institute (NCI) guidelines, compounds/extracts/fractions can be considered cytotoxic if their IC50 values are below 30 µg/mL [34,35]. Thus, based on this criterion, it was concluded that IsoP was active against Saos-2 cells (IC50 < 10 µg/mL). In the case of HT29 and multiple myeloma cells, IsoP had rather low cytotoxicity (IC50 < 30 µg/mL) and was non-cytotoxic against HOS and SW620 as the IC50 values were outside the above range. However, the significant growth inhibition of the non-sensitive type of cancer cells was due to the high concentration of the compound rather than its specific activity.

Moreover, the cytotoxic profile of IsoP was assessed against normal human skin fibroblasts (HSFs) to calculate the selectivity index (SI), which represents the mean ratio of IC50 in a normal cell culture to the mean IC50 in a particular cancer cell culture and measures the window between the cytotoxicity and anticancer activity of a given drug [36]. SI > 1.0 indicates a drug’s efficacy against cancer cells is greater than its toxicity against normal cells [37].

As shown in Figure 8A, isopimpinellin affected the proliferation of normal HSF lines to a lesser extent than the examined cancer cell lines (except SW620). An obvious decrease in cell proliferation in the HSF culture was observed after exposure to the relatively high doses of the compound (200–400 µM). Moreover, the fibroblasts also exhibited poor sensitivity to this substance when incubated (for 48 h) in the culture medium with a minimal concentration of FBS (1%) (Figure 8B). The non-linear regression analysis of the dose-response curves (Figure 8C) highlighted that IsoP was characterized by a high IC50 value of 410.7, which was approximately 4–10 times higher compared to the four cancer cell lines.

Additional data presented in Table 5 provides evidence that this furanocoumarin has a high SI value, close to 10 (SI = 9.62) in the Saos-2 cell line and close to 5 (SI = 4.88) in the U266 and HT-29 cell lines, further demonstrating its therapeutic potential for a variety of malignant tumor types.

Thus, isopimpinellin not only had no inhibitory impact on normal cell growth, but it also had no ability to kill non-cancerous cells (the characteristics of the tested cell lines are presented in Appendix A). These data support the evidence that isopimpinellin meets two important criteria for an effective chemo-preventive agent: tumor specificity and low toxicity to normal cells.

### 2.5. Impact of IsoP on DNA Synthesis and Apoptotic Cell Death

Since isopimpinellin showed the greatest growth inhibitory potential and selectivity against the Saos-2 line, these cells were selected for the further study to explore the anticancer mechanism of its action. The anti-proliferative effect of IsoP against the Saos-2 cells was a consequence of impaired cell division, as indicated by the BrdU proliferation assay. As shown in Figure 9, the 72-hour incubation with IsoP in a dose-dependent manner reduced the number of cells undergoing DNA replication. The lowest concentration at which IsoP significantly suppressed DNA synthesis was 10 µM. After the treatment with its maximal concentration (160 µM), the mitotic activity of the cells was reduced by 84%.

To test whether the cytotoxicity of IsoP could also result from the induction of cell death, the Saos-2 cells were treated with different IsoP concentrations for 72 h, and then necrotic and apoptotic cells (early and late) were measured cytometrically. IsoP at concentrations up to 40 µM resulted in only a slight (up to 4–5%) increase in necrotic cells compared to the control. However, a substantial increase in necrosis (up to 18%) was elicited by the dose of 80 µM. In contrast to necrosis, the level of apoptosis increased gradually with the increasing IsoP concentration (Figure 10A). The percentage of cells undergoing apoptosis increased from 5.52 in the control to 21.38%, 27.12%, and 47.46% at 20, 40, or 80 µM, respectively (Figure 10B).

The proapoptotic potential of IsoP was further determined by measuring active caspase-3. Caspase-3 is an executive cysteine aspartate specific proteinase activated by both the intrinsic and extrinsic death pathways in apoptosis. The activated enzyme cleaves intracellular structural proteins and functional proteins and leads to cell destruction through the mechanism of apoptosis [38]. As shown in Figure 11A,B, IsoP caused a rise in the number of osteosarcoma cells with active caspase-3 in a concentration-dependent manner. Compared to the control culture (1.43%), the amount of caspase-3 positive cells at 10 µM, 20 µM, 40 µM, and 100 µM IsoP concentrations was 2.70%, 6.71%, 12.42%, and 18.68%, respectively. The flow cytometry data clearly indicate that IsoP-mediated apoptosis is a caspase-dependent process involved in the cytotoxic effects of the tested compound.

## 3. Discussion

Cancer is a complex disease with a genetic and environmental background, and the increase in the number of cancer incidences is being estimated to reach c.a. 20 million per year by 2024 [39], which makes the disease a serious health problem in the world. To try to manage this problem, multi-targeted strategies have been developed, combining surgery, radiotherapy, and chemotherapy with synthetic drugs often co-administered with natural-derived bioactive molecules used as adjuvants or chemo-preventive agents [1,36]. For this reason, the isolation and purification of natural compounds, including furanocoumarins, from plant sources is of paramount importance. Therefore, eco-friendly methodologies, such as centrifugal partition chromatography (CPC), have recently been welcomed, as they are effective and low-cost techniques that can be additionally scaled up from the analytical to the preparative process. One of the furanocoumarins, isopimpinellin, was isolated with the use of CPC in our previous study [12]. In the present work, we evaluated *Ammi majus* L. fruits as a source of furanocoumarins, especially IsoP, which was isolated and quantitatively analyzed in extracts obtained using PLE/ASE. Since pressurized liquid extraction was previously shown to give the highest yield of simple coumarins and furanocoumarins from plants from the Apiaceae family [7,20,40,41,42], this eco-friendly technique (less time, less energy, and a solvent-consuming procedure) was used to find the optimal conditions for the extraction of target compounds. Based on conventional heating in an oven at elevated temperatures (normally exceeding the boiling point of the extracting solvent) and at pressures up to 200 bar, this dynamic extraction requires a very a short time and a small volume of the solvent and ensures complete extraction in short time with a small volume of an organic solvent due to the better penetration of the sample [7,20,40,41,42]. This extraction methodology from the *A. majus* fruit matrix was applied and optimized for the first time in our study. Previously, Królicka et al. [20] applied ASE extraction for the isolation of umbelliferon and bergapten from callus cultures of *A. majus*, and the obtained extracts were analyzed by HPLC. In our study, it was found that the best quantitative analysis was achieved using the methanol ASE extract obtained at 130 °C (ASE/Mex/130). In this extract, the content of the main furanocoumarins was quantitatively (SPE/RP-HPLC/DAD) evaluated and expressed in mg/100 g of the dry weight (DW) of the sample. The total amount of the detected compounds identified as coumarins was considered to be 100%. IsoP was the main compound in this extract, constituting 24.56% of the total calculated coumarin fraction (1.65 g/100 g). The amount of IsoP was 404.14 mg/100 g DW. The optimization of the furanocoumarin isolation from the plant material from the Apiaceae family from the *Peucedanum* or *Heracleum* genera was performed previously, and the extraction methods used included Soxhlet extraction, ultrasound assisted extraction (UAE), microwave-assisted extraction (MAE), and pressurized liquid extraction/accelerated solvent extraction (PLE/ASE); and in this case, temperatures of 110 °C and 130 °C were found to be the most suitable for the process [7,40,41,42]. In our experiment on the *Ammi majus* fruit matrix, the highest extraction yield (28.81 ± 0.49%) was obtained for the ASE/Mex/130 extract. These conditions were beneficial for the isolation of three main furanocoumarins from *A. majus* fruits, IsoP, 5 MOP, and 8 MOP, and for the quantitative estimation of the coumarin content in this plant material. However, the dichloromethane extract was also effective (18.43 ± 0.67%), and the use of this solvent enabled us to avoid extraction of polar compounds found in the methanol extract; therefore, ASE/Dex/130 was selected for preparative isolation of IsoP with the use of the LC/CPC methodology optimized previously [12]. The isolated IsoP was then the subject of biological studies presented in this paper, evaluating the chemo-preventive potential of the isolated furanocoumarin against selected tumor cell lines.

Chemotherapy and radiotherapy, i.e., the basic methods used in the fight against malignant tumors, cause serious and often long-lasting side effects in the body [43]. Another equally important problem is the ability of cancer cells to develop mechanisms enabling them to survive during cancer treatment and develop resistance to the therapy [44]. Therefore, it is extremely important to search for new bioactive substances of plant origin that can increase the effectiveness and reduce the systemic toxicity observed during anticancer treatment. Therefore, in cancer therapy, attention is paid to natural substances [45,46,47,48,49,50], and numerous studies have confirmed the effectiveness of furanocoumarins in anticancer therapy, including multidrug-resistant (MDR) cancers, for which no ideal treatment is available at present [46,51,52]. For example, bergapten and xanthotoxin were found to have cytotoxicity against gastric (EPG85.257RDB), ovarian (A2780RCIS), and breast cancer (MCF7MX) cell lines overexpressing multidrug-resistant proteins (MDR1, MRP2, and BCRP) [53]. In addition to the rapid absorption of furanocoumarins from food into the bloodstream, the effect of these substances on increasing the bioavailability of conventional anticancer drugs has also been noticed, which additionally makes them promising oncological agents acting as adjuvants [54,55,56,57,58]. As confirmed experimentally, the antitumor properties of various furanocoumarins are determined by the coexistence of both the coumarin core and the furan ring in their chemical structure, and the location, type, and number of attached substituents determine the solubility, bioavailability, direction, and strength of their pharmacological action [46,59].

The chemo-preventive ability of natural furanocoumarins has been observed in leukemia, glioma, ovarian, prostate, breast, lung, colon, and cervical cancer. Apoptosis and autophagy, anti-inflammatory and antioxidant properties, inactivation of NF-κB (nuclear factor Kappa-light-chain-enhancer of activated B cells), inhibition of PI3K/Akt (phosphatidylinositol 3-kinase/serine/threonine kinase), cell cycle inhibition ability, and modulation of p53 protein have been confirmed as important for the activity of these compounds [30,31,46,59].

In recent years, a number of biological activities of furanocoumarins (including xanthotoxin, bergapten, bergaptol, angelicin, bergamottin, and imperatorin) have been confirmed, indicating their suppressor potential in relation to various models of cancer cells [60,61]. Also, the recently presented results of research conducted in our group provided evidence for the antiproliferative, antimigratory, and proapoptotic effects of xanthotoxin and bergapten (not activated by UV radiation), depending on the concentration of these compounds and the type of cancer cells [29,30]. Individual preliminary studies have drawn our attention to a structurally similar furanocoumarin, isopimpinellin, as a promising tumor growth inhibitor with antiangiogenic activity [52,62,63,64,65]. The lack of detailed data on the anticancer potential of isopimpinellin against various types of cancer in in vitro and in vivo studies indicates an urgent need to continue research to further verify its anticancer effect. Our research in this area started by assessing the ability of isopimpinellin to disrupt the proliferation state of cultured osteosarcoma cells (HOS, Saos-2), colorectal cancer cells (HT-29, SW620), and multiple myeloma cells (RPMI8226, U266).

In our experiments, the potential toxicity of the tested isopimpinellin towards dividing normal human fibroblasts (HSFs) was also checked, and the obtained IC50 value was 410.7 µM, confirming the lack of toxicity of IsoP in the range of the tested concentrations. Previous studies conducted by other research teams provided the first evidence that IsoP works against certain types of cancer cells. For example, in cultures of human metastatic melanoma cells (FM55M2 and FM55P), the IC50 values for IsoP ranged from 129.4 µM to 156.8 µM, respectively [66]. In the case of SW480 colon cancer cells [67], only a very high IsoP concentration of 200 µM impaired their cellular growth. Our experimental data from the MTT assay confirmed the differential dose- and cell-type-dependent cytotoxic effects of the tested furanocoumarin. The comparison performed using IC50 showed that the most resistant cell culture to IsoP was SW620 (IC50 = 711.3 µM), while the most responsive culture was Saos-2, with IC50 = 40.05 µM, i.e., the lowest value. In turn, the U266, HT29, RPMI8826, and HOS lines (with IC50s of 84.16, 95.53, 105.0, and 321.6 µM, respectively) were much less sensitive or insensitive. Interestingly, as observed in our previous studies of bergapten (5 MOP, with a functional methoxyl group -OCH3 at the C5 position in the molecule), the highest level of inhibition of mitochondrial metabolism was demonstrated in cell lines derived from primary cells and non-metastatic tumors [31]. Bergapten showed greater activity against primary bone and colon tumors (Saos-2 and HT-29, respectively) compared to their invasive or metastatic counterparts (HOS and SW620, respectively) [31]. These tendencies were also analyzed and observed in the case of colorectal carcinomas by other authors [68]. A similar trend was observed in the case of IsoP (with two methoxy groups at C5 and C8) tested in the present study (greater activity against Saos-2 and lower activity against SW620 cells). Moreover, a study of IsoP conducted by Patil et al. on the SW480 line (derived from a grade between III and IV colon adenocarcinoma) showed a 30% decrease in cell proliferation induced by IsoP, but only after exposure to such a concentration of IsoP as 200 µM, while lower concentrations did not show any toxicity [67]. In contrast, xanthotoxin (8 MOP; meth-oxy group at C8 position), previously studied in our experiments conducted on the colon cancer cell lines, proved to be an almost two-times more potent inhibitor of the growth of SW620 cells (IC50 = 88.5 µM) than HT-29 (IC50 = 159.5 µM) [30]. Consistent with previous reports [46,49,59,68,69,70], we indicated that the differential potential of linear furanocoumarins in the same cancer cell lines is largely due to the differences in the chemical structure of these compounds (such as the different location, presence, or absence of additional substituents attached to the coumarin core). Our observations suggest that the presence of an additional methoxy group at the C8 position in the furanocoumarin molecule may be important for increasing the reactivity of some cancer cells to the action of these compounds. In our previous experiments performed on the RPMI8226 and U266 cell lines, only a minimal antiproliferative response was obtained after 96-hour incubation with monosubstituted furanocoumarins, such as 5 MOP and 8 MOP (in the case of 5 MOP, 1272 µM and 1190 µM, respectively, and in the case of 8 MOP, in U266 = 309.3 µM) [30,31]. Compared to these compounds, IsoP (with two methoxy groups at the C5 and C8 positions) showed a much stronger effect on the proliferation of both of these cell lines, which was confirmed in our study (IC50 84.16 and 105.0 µM, respectively). Yang et al. [70] demonstrated high activity of IsoP isolated from the fruits of *Cnidium monnieri* L. against human acute promyelocytic leukemia cells (HL-60), with an IC50 value of not more than 50 µM. A similar IC50 value for the above cells was presented by Kubrak et al. [52]. Interestingly, in their research, in addition to HL-60, other leukemic cells with a resistance phenotype were found to be equally or even slightly more susceptible to the treatment [52]. Despite extensive research on various compounds from the psoralen group, there are still few data on the anticarcinogenic mechanism of IsoP action in both in vitro and in vivo systems. Isopimpinellin was documented to inhibit chemically induced skin tumors in mice by blocking the formation of DNA adducts and impairing the activation of CYP1 and CYP2B involved in the metabolism of carcinogens [62,63,64]. As mentioned previously, IsoP has an inhibitory impact on the angiogenesis process, which was reported by Bhagavatheeswaran et al., confirming the ability of IsoP to inhibit proliferation, migration, and the invasion of endothelial cells (HUVEC) and to modulate several pro-angiogenic (VEGF, AKT, and HIF-1α) and angiostatic (miR-542-3p) markers. Thus, isoP may prevent the spread of cancer cells in the human body. The antiangiogenic effect of isopimpinellin was also confirmed in vivo using a zebrafish model [65]. Our mechanistic study confirmed that the cytotoxic effects of IsoP on sensitive Saos-2 cells were related to stimulation of the apoptosis pathway via activation of central suicide caspase 3. In addition to apoptosis, the IsoP exposure also caused a concentration-dependent decrease in the incorporation of labeled BrdU during DNA replication, indicating the ability of this molecule to disrupt osteosarcoma cell division. Similarly, most structural analogues of furanocoumarins have been found to exert cytotoxic effects by inducing apoptosis and inhibiting the cell cycle. The previously tested bergapten (5 MOP) was able to induce cell cycle arrest in the G2 phase, deregulate the pro-apoptotic expression of Bax and the anti-apoptotic expression of Bcl-2, and stimulate the activity of caspase 9 [31]. In studies of neuroblastoma cell lines (SK-N-AS and SW620), another linear furanocoumarin xanthotoxin (8 MOP) was found to block DNA replication and trigger apoptotic cell death through both extrinsic and intrinsic apoptosis pathways and downregulation of pro-survival PI3K/AKT signaling [30]. The same mode of its pro-apoptotic action was observed in HepG2 cells in other studies [69,70]. Moreover, certain furanocoumarins were found to exhibit anti-invasive behavior [50] and/or anti-inflammatory properties. In their studies, Robertson et al. [29] confirmed that IsoP induced neutrophil apoptosis, influencing caspase-3 activity in a zebrafish in vivo model of inflammation. It was found that isopimpinellin affected PI3K activity and, consequently, neutrophil motility during the recruitment phase of inflammation. Additionally, some furanocoumarins have been reported to have chemotherapeutic synergistic potential when applied in combination with other anti-tumor drugs [45,46].

The question of whether IsoP has strong anti-migratory and anti-metastatic properties remains to be resolved. Clarification of the main issues related to the various manifestations of its anti-cancer effects will allow scientists to take a better look at this furanocoumarin as a promising molecule in the fight against cancer.

It is well known that, in addition to the ability to destroy cancer cells, a good candidate for an anticancer drug should be characterized by no (preferably) or low toxicity towards normal cells [36,37], and the above features are a priority in the development of various therapeutic regimens. The literature data regarding the pharmacological profile of isopimpinellin are limited; however, they indicate low or no systemic in vivo toxicity of the compound in mice, even when administered in high doses [63]. Our in vitro study also showed both no lethal and antiproliferative effect of IsoP on normal HSF cells over a wide range of concentrations, indicating better tolerance of these cells (IC_50_ = 410.70 µM) compared to the most tested cancer cell lines. Despite the interesting results obtained, there is still a recommendation to conduct further studies (in vitro and in vivo models) to elucidate the molecular and cellular mechanisms of IsoP action, in particular against the most sensitive types of cancer cells. In the near future, we plan to expand the research towards anti-metastatic and anti-migratory assessment and the impact of IsoP on key signaling pathways related to growth, survival, and epithelial-mesenchymal transition (EMT) markers (such as vimentin, fibronectin, N-cadherin, MMP-2, and MMP-9 important in the initiation of tumor metastasis) to better understand the biological potential of this promising natural compound, and these investigations are in progress.

## 4. Materials and Methods

### 4.1. Solvents and Chemicals

Acetonitrile and methanol (HPLC gradient grade) were purchased from J.T. Baker (Deventer, The Netherlands); *n*-hexane, dichloromethane, ethyl acetate, and petroleum ether (PE) were of an analytical grade (POCH (Gliwice, Poland). Ultrapure water (18.2 MΩ), obtained from a Simplicity (Millipore, Molsheim, France) purification system, was used. For LC-MS experiments, the water and acetonitrile were of LC-MS grade (J.T. Baker, Deventer, The Netherlands). Umbelliferon (UM), xanthotoxin (8 MOP), isopimpinellin (IsoP), bergapten (5 MOP), and imperatorin (IMP), all of purity ≥ 98% (assayed by HPLC, Sigma, St. Louis, MO, USA), were used as reference substances.

### 4.2. Plant Material

Mature fruits of *Ammi majus* L. (Apiaceae, A/AM-1/2012) were collected in the Pharmacognostic Garden of the Medical University of Lublin (51°25′ N, 22°56′ E, 200 m AMSL, Lublin, Poland) and identified by M. Bartnik, PhD. Before use, fruits were dried in a dark place at room temperature (humidity < 30%, temp. c.a. 25 °C).

### 4.3. Isolation, Purification, and Identification of IsoP

#### 4.3.1. Extraction of the Plant Material and Purification of the Samples

Dried plant materials were air-dried, ground to powder, sieved as recommended by Ph. Eur. 10th Ed. [71], and kept in the glass container in a dark place until analysis.

For the quantitative analysis, PLE/ASE (pressurized solvent extraction/accelerated solvent extraction) was performed using the Dionex ASE-100 apparatus (Sunnyvale, CA, USA). The dried plant material was extracted by dichloromethane (CH_2_Cl_2_; Dex), and methanol (MeOH; Mex), respectively. Each time (*n* = 4), 0.5000 g ± 0.0001 of dried plant material was used, and the PLE/ASE extraction parameters were as follows; 3 static cycles; cycle duration, 10 min; purge time, 100 s; and flush volume, 60% [41]. Various temperatures,—50, 70, 90, 110, and 130 °C—were tested for each solvent used.

PLE/ASE extracts (Dex and Mex) were collected, evaporated under vacuum, and carefully weighed to calculate the extraction yield, as follows;
$$extraction\;yield\;[\%]=\frac{obtained\;extract\;[mg]}{plant\;substance\;[mg]}\times 100\%$$

The obtained extracts were dissolved in the methanol and placed in the volumetric flasks (25 mL). Samples (5 mL) were purified by SPE, as described previously [41], before HPLC/DAD and ESI-TOF MS analyses.

For preparative purposes, pulverized plant material was extracted with dichloromethane by use of PLE/ASE (3 × 10 g), and the combined ASE/Dex/130 extract was concentrated under vacuum, placed in the refrigerator for 7 days, and the collected semicrystalline coumarin sediment (SCS = 2.12 g) was subjected to preparative isolation by the previously optimized LC/CPC method [12].

#### 4.3.2. HPLC Method Validation

For the quantitative analysis, the HPLC method was validated, and selectivity, precision, repeatability, linearity, and recovery were analyzed. Selectivity was confirmed by the DAD spectra of the compounds collected in the range of 190–400 nm. The analysis was monitored in each case online by DAD at 254 nm, 320 nm for coumarin compounds, and 360 nm for impurities. The precision was measured by the relative standard deviations (RSD) of peak areas for each reference standard, and six replicates from the same sample of *Ammi majus* fruits (ASE/Mex/130) were tested for the repeatability of the method.

All of the quantitative calculations were performed with external standards, and the stock solutions for the analyzed coumarins were as follows: UM and IsoP 1 mg/10 mL; 8 MOP, 5 MOP, and IMP 2.2 mg/10 mL.

A series of the working standard solutions containing tested coumarins were prepared by diluting the stock solutions within the concentration range of 1–100 µg/mL for UM, 7–220 µg/mL for 8 MOP, 5–100 µg/mL for IsoP, and 1–220 µg/mL for 5 MOP and IMP, with methanol, and their peak areas were measured (*n* = 3) at λ = 320 nm. Linear least squares regression of the peak areas as a function of the concentrations was performed to determine the correlation coefficients (R^2^). The limit of detection (LOD) and limit of quantification (LOQ) were calculated from the calibration plots for the main coumarins detected in the extract (UM, 8 MOP, IsoP, 5 MOP, and IMP) by applying the following formulas: LOD = 3.3*δa*^−1^ and LOQ = 10*δa*
^−1^ (where *δ* is the standard deviation of the response, and *a* is the slope of the calibration curve).

The accuracy was tested as recovery tests of the SPE purification step, as described previously [41,72]. For this purpose, standards of coumarins (IsoP, 5 MOP, and IMP; 1 mg/10 mL of methanol), and the spiked ASE/Mex/130 extract (5 mL of the extract combined with 5 mL of the standard solution) were analyzed.
$$recovery\;[\%]=\frac{total\;detected\;amount-original\;amount\;[µg]}{spiked\;amount\;[µg]}\times 100\%$$

The percentages of the mean recovery were as follows: 100.4 ± 0.7 and 99.8 ± 0.5 for IsoP, 99.8 ± 1.7 and 98.9 ± 0.4 for 5 MOP, and 101.4 ± 0.9 and 99.5 ± 0.7 for IMP, for tested standards and spiked samples, respectively. The recoveries of the spiked samples were lower than in the case of the standards, indicating the influence, however, not highly significant, of the plant matrix (ASE/Mex/130). The relative standard deviation (RSD (%) = (SD/mean) x 100) values were in each case less than 1.8%.

The HPLC equipment used was an Agilent 1100 system (G1311A quaternary pump, a thermostat ALS Therm G1330B, G1315B DAD, ALS G1329 autosampler, and a G1322 membrane degasser) with the injection valve Rheodyne with a 20 µL sample loop, controlled by Agilent HPLC OpenLab CDS ChemStation software v. 2. (Agilent Technologies, Santa Clara, CA, USA). The Agilent chromatograph was equipped with the Hypersil ODS C18 column (250 × 4.4 mm ID, 5 µm; Agilent Technologies, Santa Clara, CA, USA). The flow rate was 1 mL/min, temperature 25 °C, and the mobile phase gradient elution (methanol-A in water-B) was as follows: 0–5 min; 40% A in B; 5–10 min; from 40 to 60% A in B; 10–20 min; 60–80% A in B, 20–30 min; 80–85% A in B, post-time 10 min.

The coumarin content was expressed in mg/100 g (and in %) of the dry weight of the plant material, and the total coumarins detected in each extract (sum of those possessing umbelliferon-like, xanthotoxin-like, bergapten-like, and imperatorin-like spectra) were considered to be 100%. The amount of the key furanocoumarins (IsoP, 8 MOP, and 5 MOP) in the obtained PLE/ASE extracts was estimated.

#### 4.3.3. LC/CPC Isolation of Isopimpinellin

Isolation of IsoP was performed from the SCS obtained from the ASE/Dex/130 extract. The SCS dissolved in the dichloromethane was mixed with the stationary phase (silica gel Si60 230-400 mesh, Merck, Germany), and the solvent was evaporated under a vacuum. Next, SCS was separated on a preparative LC silica gel column in a gradient of ethyl acetate in dichloromethane (0–80%; *v*/*v*), and the first 10 fractions (50 mL of each) were collected. Fraction FR6, containing the targeted IsoP, was evaporated, dissolved in the mixture of two CPC phases, and separated by the optimized centrifugal partition chromatography (CPC), as was described previously [12]. The CPC instrument employed in the present study was a model Armen SCPC-250-L (Armen Instrument, Saint Ave, France) equipped with an SCPC-250 Teflon column with a total capacity of 250 mL integrated with a gradient flow pump and a UV lamp (Flash 06S DAD 600) operating at various wavelengths and equipped with a manual injector with a 10 mL sample loop. The system was controlled by the software Armen Glider CPC V5.0a.05.

In each separation run, the column was first filled entirely with the stationary phase at a flow rate of 20 mL/min (500 rpm, 20 min.). Then, the upper organic phase was pumped into the column. After the equilibration of the column, 40 mg of the SCS (dissolved in the mixed two phases (4 mL of each, finally 8 mL) was injected through the injection valve. An effluent from the column was continuously monitored by UV detection (at 254 nm and 320 nm, suitable for coumarin detection), and the peak fractions (18 mL of each) were collected by an automatic fraction collector. The separation was performed in the ascending mode with the use of the optimized biphasic solvent system (*n*-hexane/ethyl acetate/methanol/water; 10:8:10:9, *v*/*v*/*v*/*v*). The flow rate was 3 mL/min, and the rotation speed was 1600 rpm [12].

#### 4.3.4. ESI-TOF MS Identification of Isopimpinellin

The identity of isolated isopimpinellin was confirmed additionally in MS experiments performed with the use of a 6210 ESI-TOF MS mass spectrometer (Agilent Technologies, Santa Clara, CA, USA), equipped with a G1312A quaternary pump, a thermostat TCC SLG1316B, G1315B DAD, ALS G1329A autosampler, and a G1379B membrane degasser, and with Mass Hunter Software v. 10.0. The stationary phase was an XBridge^TM^ Shield RP18 (100 × 2.1 mm ID, 3.5 µm) column (Waters; Milford, MA, USA). The mobile phase (flow 0.2 mL/min; pH 4.5; HCOONH_4_) was composed of 1% MeCN (A) and B; 95% MeCN (B), and the gradient elution was as follows: 1–15 min 30% B; 15–25 min 45% B; 25–35 min 85% B, and the post time took 10 min. The analysis was performed in the mass range of 100–1000 *m/z*. Fragmentor 215 V (positive ion mode), capillary voltage 4000 V, skimmer 65 V, nebulizer pressure 35 psi, gas temperature 350 °C, nitrogen flow 10 L/min, and in each case, 2 µL of the sample was injected.

### 4.4. Biological Studies

#### 4.4.1. Cell Lines

Established human cancer cell lines derived from primary and metastatic osteosarcoma (Saos-2 and HOS), primary and metastatic colorectal adenocarcinoma (HT-29 and SW620), and multiple myeloma (RPMI8226, U266) were used to evaluate the in vitro anticancer potential of isopimpinellin. The characteristics of the aforementioned lines are shown in Appendix A. These cell lines (except HT-29) were purchased from ATCC, Manassas, VA. The HT-29 cell line was obtained from the Institute of Immunology and Experimental Therapy (Polish Academy of Sciences, Wroclaw, Poland). Normal human skin fibroblasts (HSFs) were a laboratory strain established using an outgrowth technique from the skin explants of healthy subjects (with written informed consent). The U266 and RPMI8226 cells were grown in an RPMI1640 medium, the Saos-2 and HT-29 cells in McCoy’s 5A Modified Medium, SW620 in an L-15 medium, and HOS and HSF in a MEM medium. All of the media purchased from Sigma Aldrich (St. Louis, MO, USA) were supplemented with 10% fetal bovine serum (PAN-Biotech GmbH, Aidenbach, Germany) and antibiotics such as penicillin (100 IU/mL) and streptomycin (100 µg/mL) (Sigma-Aldrich). Cultures were kept at 37 °C in a humidified atmosphere of 95% air and 5% CO_2_.

Stock solutions of the isolated isopimpinellin were prepared in DMSO (Sigma-Aldrich) and stored at 4 °C until use. The DMSO concentration did not exceed 0.1% (*v*/*v*). The compound was diluted in an appropriate culture medium (supplemented with 1% or 10% FBS) immediately before use.

#### 4.4.2. Proliferation MTT Assay

The antiproliferative potential of isopimpinellin against cancer cells was evaluated after 96 h exposure using the colorimetric MTT assay, and the IC50 values were calculated as described previously [70,73]. Cancer cells were plated on a 96-well culture plate (Nunc, Rochester, NY, USA) at a density of 2 × 10^5^ (RPMI8226, U266), 3 × 10^4^ (HSF, HT-29, Saos-2), and 1 × 10^4^ (SW620, HOS). The cells (except RPMI8226 and U266) were additionally incubated for 24 h to cell attachment. Next, 100 µL or 50 µL (in the case of MM cells) of serial dilutions of isopimpinellin prepared in a fresh medium (10% FBS) were poured into the appropriate wells (the final concentrations of the compound were 3.125, 6.25, 12.5, 25, 50, 100, and 200 µM). The cells were then cultivated for 4 days, followed by an addition of 15 µL of the MTT solution (5 mg/mL, Sigma Chemicals) into each well and an additional incubation of 3 h. The blue formazan crystals were then dissolved overnight in an SDS buffer, and the colored product was quantified spectrophotometrically by measuring absorbance at 570 nm using an EL800 Universal Plate Reader (Bio-Tek Instruments, Inc., Santa Clara CA, USA). The absorbance of the control wells was taken as 100%, and the results were expressed as a percentage of the control.

#### 4.4.3. Cytotoxicity Assessment—NR Assay

The cytotoxicity assay was performed to determine the toxic doses against normal HSF cells. The cells were seeded in 96-well plates and allowed to adhere overnight. On the following day, the growth medium was removed, and the cells were exposed to serial dilutions (of isopimpinellin 3.125–200 µM) in a fresh medium with 1% FBS (growth restriction). After 48 h, the cell toxicity was determined using the neutral red (NR) uptake assay as described previously [31].

#### 4.4.4. Selectivity Index (SI)

To determine the cytotoxic selectivity of the IsoP, the selectivity index (SI) was calculated according to the following equation:SI = mean IC50 non-cancer cells/mean IC50 cancer cells

The IC50 for normal cells should be higher than the IC50 for cancer cells, suggesting that cancer cells are killed before normal cells. SI ≥ 10 was considered to belong to a selective compound [36,37].

#### 4.4.5. BrdU ELISA Cell Proliferation Assay

The Saos-2 cancer cells were placed on 96-well plates (Nunc) at a density of 3 x 10^4^cells/mL. After 24 h, the cells were exposed to 5–160 µM of isopimpinellin for 72 h. DNA synthesis in the dividing cells was estimated by measuring 5-bromo-20-deoxyuridine (BrdU) incorporation using a Cell Proliferation ELISA, BrdU kit (Roche; Basel, Switzerland), according to the manufacturer’s instructions. The absorbance values were measured at 450 nm using a microplate reader.

#### 4.4.6. Flow Cytometry

The Saos-2 cancer cells were placed on 6-well plates (Nunc) at a density of 3 × 10^5^ cells/mL. On the following day, the culture growth medium was removed, and the cells were treated with a selected concentration of (10–80 µM) isopimpinellin for 72 h. The apoptotic and necrotic cell populations in control and treated cell cultures were examined in the flow cytometer (BD FACS Calibur) using the Propidium Iodide/AnnexinV-FITC method, according to the manufacturer’s instructions. The fluorescence-activated cell sorting (FACS) technique was also employed to assess the active form of caspase-3. After 72 h of exposure to IsoP, the FITC Active Caspase-3 Apoptosis Kit (BD Biosciences, San Jose, CA, USA) was used according to the manufacturer’s instructions. All experiments were performed in triplicate and gave similar results.

### 4.5. Statistical Analysis

In phytochemical quantitative studies, a statistical analysis was performed using a students’ *t*-test, *p* ≤ 0,05; *n* = 6, and the experimental data were presented as the mean ± SD. In biological studies, the experimental data were presented as the mean ± SD for at least the triplicate determination of three independent experiments. The data were analyzed by a one-way ANOVA followed by Dunnett’s test for multiple comparisons between pairs. *p* values of <0.05 were considered statistically significant. For non-linear regression analysis, the concentrations used were transformed to a logarithmic scale to calculate the IC50 values. GraphPad Prism ver. 6.05 (GraphPad Software, Inc., San Diego, CA, USA) was used for these analyses.

## 5. Conclusions

In our study, we proved that the PLE/ASE extraction process is convenient for the isolation of bioactive furanocoumarins, especially isopimpinellin, from the *Ammi majus* fruit matrix for both quantitative and preparative purposes, and the LC/CPC methodology was effective in the isolation of IsoP from this plant matrix. The isolation process could be scaled up, especially when the ASE extractor and cell size are magnified, and the CPC isolation can be easily transformed from the analytical to preparative scale. However, it is preferable to try to use more eco-friendly extraction solvents, such as mixtures of ethanol and ethyl acetate, for the isolation of coumarins, and the applied methodology enables the avoidance of the tedious and solvent-consuming preparative TLC methodology previously used for the isolation of this coumarin from this plant source, making this process more eco-friendly.

In biological studies, isopimpinellin was found to be non-toxic to normal HSF cells, and their high selectivity index (close to 10) indicates that it could be a selective molecule with potential in chemo-prevention of especially primary tumors, as indicated in the case of the Saos-2 cell line (osteosarcoma). The proapoptotic activity of IsoP was confirmed by its ability to influence caspase-3 activity, and the BrdU test indicated the ability of IsoP to interfere with Saos-2 cell division. However, there is still a need to profoundly evaluate the chemo-preventive potential of isopimpinellin and study the mechanism of its action in in vitro and in vivo models in future experiments.

## Figures and Tables

**Figure 1 molecules-29-02874-f001:**
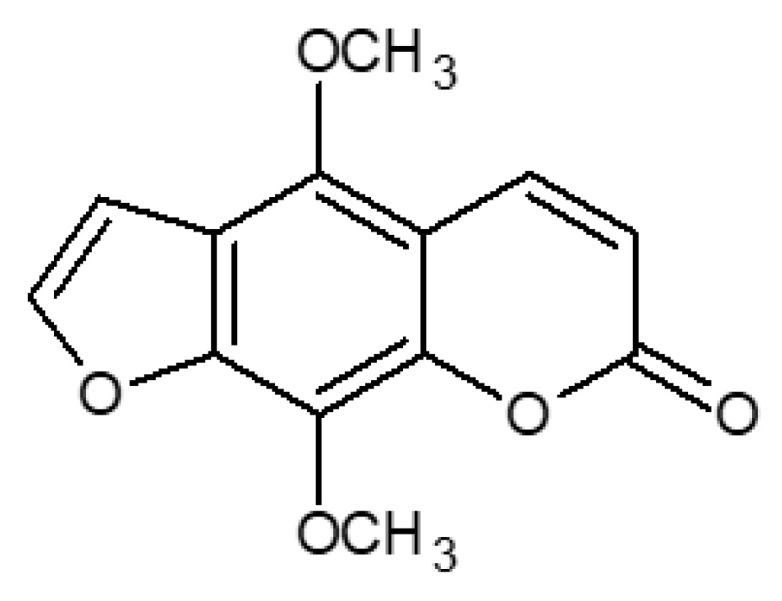
Chemical structure of isopimpinellin (5,8-methoxypsoralen; IsoP).

**Figure 2 molecules-29-02874-f002:**
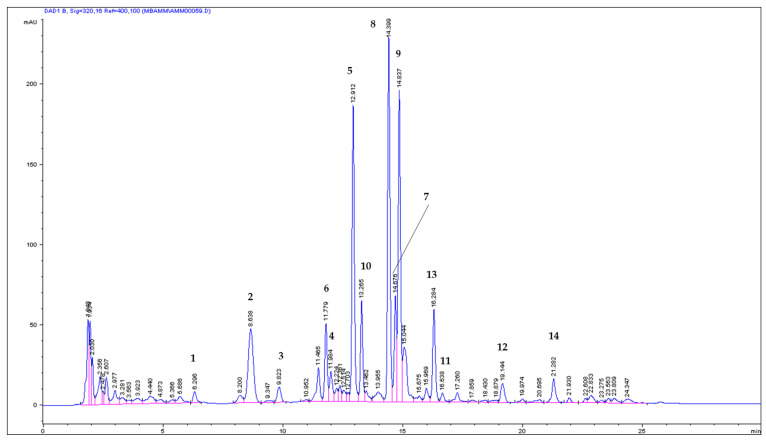
HPLC chromatogram (analyzed at 320 nm) of coumarin compounds in ASE Mex/130 from *A. majus* fruits; 1: umbelliferon; 5: xanthotoxin; 8: isopimpinellin; 9: bergapten; and 12: imperatorin. Coumarins with umbelliferon-like spectra, 2, 3, 4; with xanthotoxin-like spectra, 6, 7; with bergapten-like spectra, 10, 11; and with imperatorin-like spectra, 13, 14.

**Figure 3 molecules-29-02874-f003:**
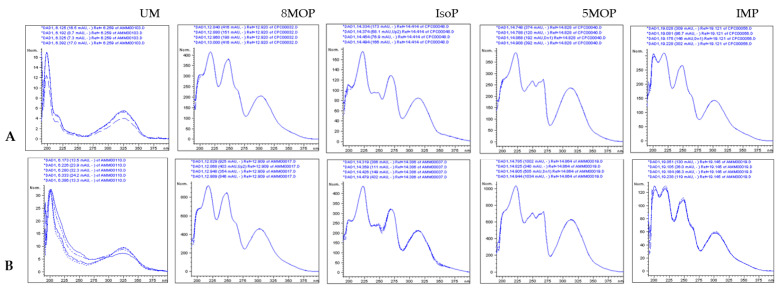
HPLC/DAD UV spectra of the standards and coumarins detected in the Mex/130 extract; (**A**) DAD/UV spectra of reference standards; (**B**) DAD/UV spectra of detected compounds; UM: umbelliferon; 8 MOP: xanthotoxin; IsoP: isopimpinellin; 5 MOP: bergapten; and IMP: imperatorin.

**Figure 4 molecules-29-02874-f004:**
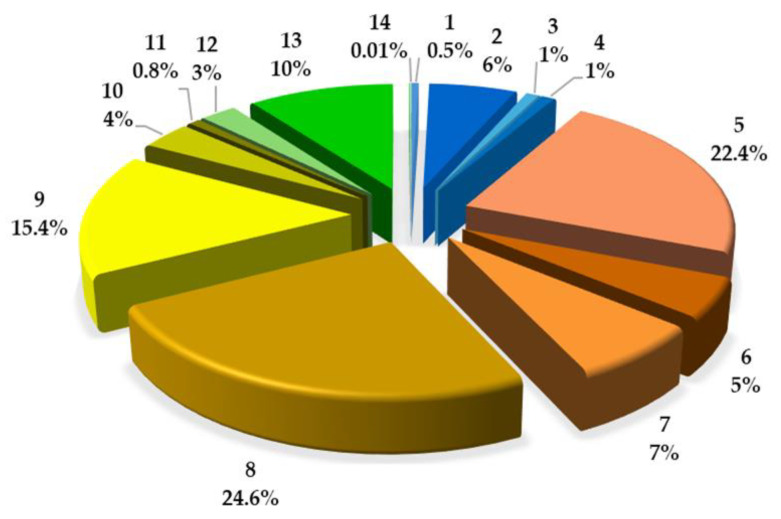
Quantitative estimation of the coumarin composition in the methanol ASE/Mex/130 extract from *A. majus* fruits. Coumarin compounds; 1: umbelliferon; 5: xanthotoxin; 8: isopimpinellin; 9: bergapten; and 12: imperatorin. Coumarins with umbelliferon-like spectra, 2, 3, 4; with xanthotoxin-like spectra, 6, 7; with bergapten-like spectra, 10, 11; and with imperatorin-like spectra, 13, 14.

**Figure 5 molecules-29-02874-f005:**
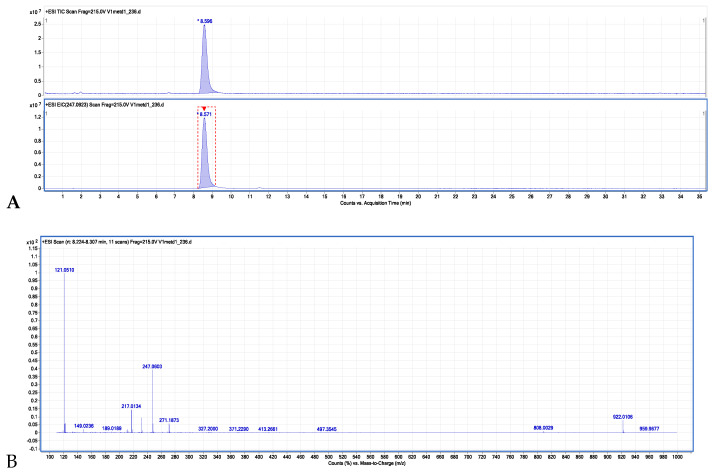
Total ion chromatogram (TIC), extracted ion chromatogram (EIC) (**A**), and the MS spectrum (**B**) of the isolated isopimpinellin (*).

**Figure 6 molecules-29-02874-f006:**
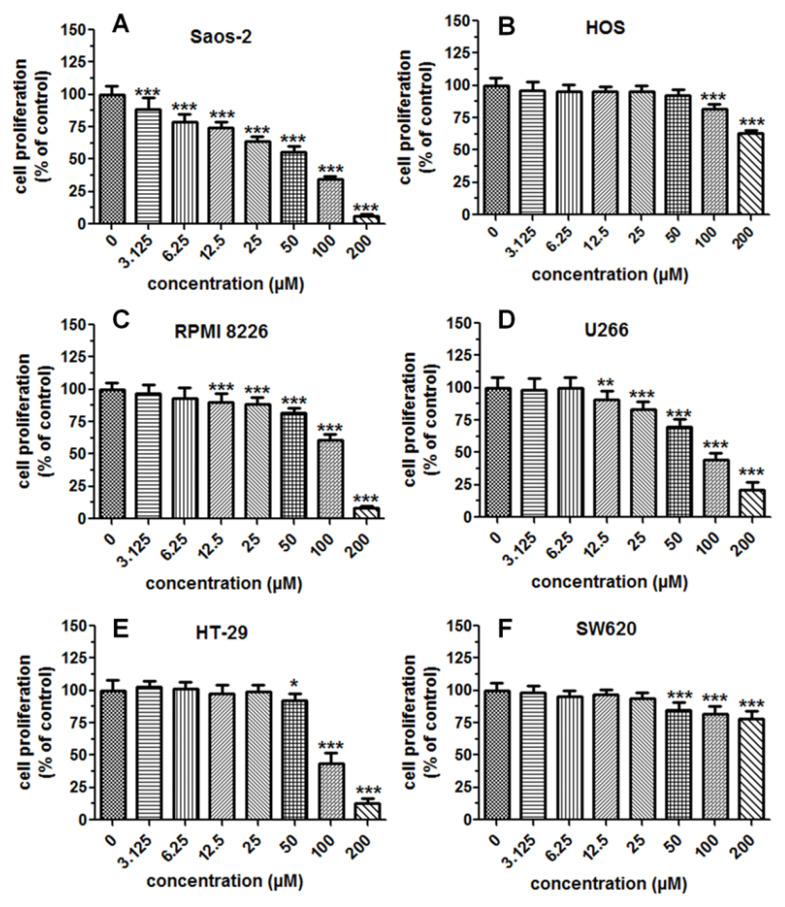
Impact of IsoP treatment on proliferation/viability of selected cancer cell models. Saos-2 (**A**), HOS (**B**), RPMI8226 (**C**), U266 (**D**), HT29 (**E**), and SW620 (**F**). The cells were cultured alone or in the presence of different concentrations of the compound, and the antiproliferative activity of IsoP was determined by the MTT assay at 96 h. Each bar represents the mean ± standard deviation (SD) of three independent experiments. * *p* < 0.05, ** *p* < 0.01, and *** *p* < 0.001 in comparison to the control (DMSO), one-way ANOVA test.

**Figure 7 molecules-29-02874-f007:**
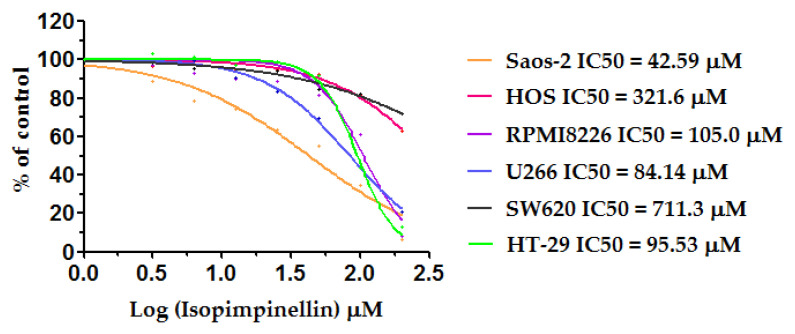
Non-linear curve fitting for the dose response curve of isopimpinellin in the Saos-2, HOS, RPMI8226, U266, SW620, and HT-29 cell lines, and their corresponding IC50 values (µM).

**Figure 8 molecules-29-02874-f008:**
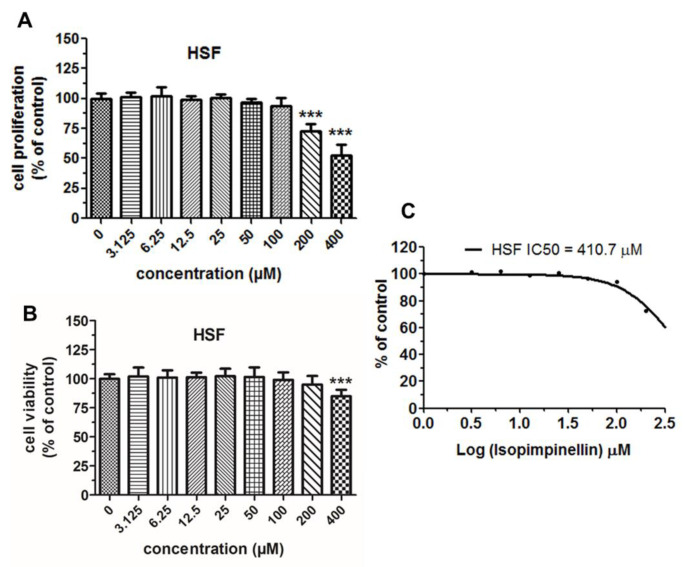
Cytostatic and cytotoxic activity of isopimpinellin (IsoP) against normal HSF cells. The cells were incubated with various concentrations of IsoP for 96 h (**A**) or 48 h (**B**) to evaluate their proliferation rate (by the MTT assay) and cell viability (by the NR assay), respectively. A nonlinear regression analysis of the average proliferation values of IsoP against HSF was used for the IC50 calculation (**C**). Each bar represents the mean ± standard deviation (SD) of three independent experiments; *** *p* < 0.001 in comparison to the control (DMSO), one-way ANOVA test.

**Figure 9 molecules-29-02874-f009:**
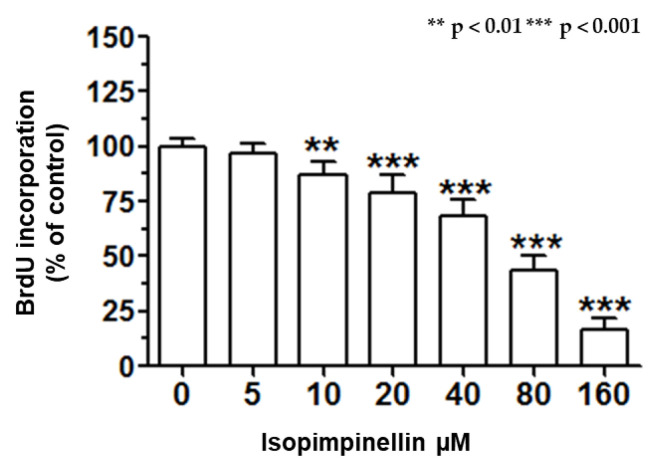
Effect of IsoP on DNA synthesis in Saos-2 cells after 72 h of treatment. BrdU incorporation was used as a marker of cell division. Each bar represents the mean ± standard deviation (SD) of three independent experiments: ** *p* < 0.01, *** *p* < 0.001 in comparison to the control (one-way ANOVA test).

**Figure 10 molecules-29-02874-f010:**
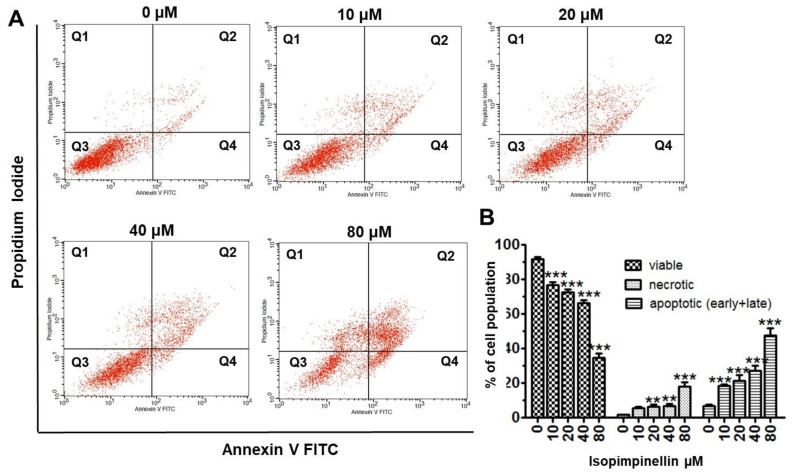
Effect of IsoP treatment on apoptotic and necrotic cell death. (**A**) Flow cytometry analysis of apoptosis and necrosis in Saos-2 cell cultures after 48-hour exposure to increasing concentrations of the compound. The apoptotic cell population in quadrants Q2 and Q4 represents early and late apoptosis. The Q1 and Q3 quadrants represent necrotic and normal cell populations, respectively. (**B**) Statistical analysis of the percentages of apoptotic and necrotic cells. Each bar represents the mean ± standard deviation (SD) of three independent experiments. ** *p* < 0.01, *** *p* < 0.001 in comparison to the control (one-way ANOVA test).

**Figure 11 molecules-29-02874-f011:**
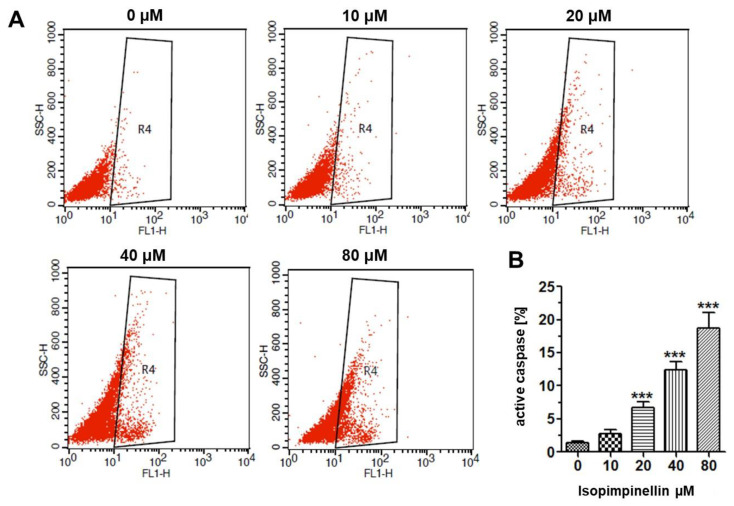
Effect of IsoP on caspase-3 activity. (**A**) Representative flow cytometry dot plot graphs of the Saos-2 cell line after treatment with IsoP for 72 h. Region R4 includes apoptotic cells with active caspase-3. (**B**) Statistical analysis of the percentages of caspase-3 activity. Each bar represents the mean ± standard deviation (SD) of three independent experiments. *** *p* < 0.001 in comparison to the control (one-way ANOVA test).

**Table 1 molecules-29-02874-t001:** Extraction yield calculated for the ASE/PLE extraction of the *Ammi majus* fruits in different solvents and at increasing temperatures.

Solvent	Temp. (°C)	ASE Extraction Yield (% ± SD) *	RSD (%)
**dichloromethane**	Dex/50	13.44 ± 0.32	2.38
	Dex/70	13.87 ± 0.17	1.23
	Dex/90	16.69 ± 0.26	1.56
	Dex/110	17.47 ± 0.31	1.77
	Dex/130	18.43 ± 0.67	3.64
**methanol**	Mex/50	18.01 ± 0.41	2.28
	Mex/70	21.60 ± 0.66	3.06
	Mex/90	24.79 ± 0.73	2.94
	Mex/110	27.09 ± 0.58	2.14
	Mex/130	28.81 ± 0.49	1.70

* *n* = 2. For the ASE extraction yield RSD ≤ 3.64. RSD (%) = (SD/mean) × 100.

**Table 2 molecules-29-02874-t002:** Validation data of HPLC/DAD analysis for coumarins present in PLE/ASE *A. majus* extracts.

Compound	t_R_ (min ± SD) *	Calibration Eq. ^&^	R^2^	Range (µg/mL)	LOD (µg/mL)	LOQ (µg/mL)
UM	6.301 ± 0.01	y = 43.468x − 1.8877	0.9999	1–100	0.02	0.07
8 MOP	12.929 ± 0.04	y = 19.618x + 59.806	0.9999	7–220	0.20	0.61
IsoP	14.419 ± 0.06	y = 23.379x + 19.515	0.9999	5–100	0.06	0.19
5 MOP	14.860 ± 0.10	y = 31.995x + 63.463	0.9996	1–220	0.42	1.26
IMP	19.189 ± 0.09	y = 14.949x + 27.773	0.9996	1–220	0.15	0.45

* *n* = 6; ^&^ *n* = 3.

**Table 3 molecules-29-02874-t003:** Quantitative results of the estimation of furanocoumarin content in the ASE/Mex/130 extract from *A. majus* L. fruits (calculated from the dry wt. of the plant material; mean ± SD).

No	Compound	mg/g (Mean ± SD)	mg/100 g	RSD (%) ^&^	(%) *	t_R_ (min)
1	UM	0.084671 ± 0.0008	8.4671	0.98	0.52	6.301
2	u	1.005768 ± 0.0739	100.5768	7.35	6.11	8.668
3	u	0.152076 ± 0.0076	15.2076	5.01	0.92	9.845
4	u	0.155196 ± 0.0025	15.5196	1.60	0.94	12.000
5	8 MOP	3.680345 ± 0.0451	368.0345	1.23	22.36	12.929
6	x	0.828362 ± 0.0170	82.8363	2.06	5.03	11.795
7	x	1.156771 ± 0.0437	115.6771	3.76	7.03	14.699
8	IsoP	4.041358 ± 0.0354	404.1358	0.88	24.56	14.419
9	5 MOP	2.530536 ± 0.0356	253.0536	1.41	15.38	14.860
10	y	0.584874 ± 0.0145	58.4874	2.48	3.55	13.282
11	y	0.130425 ± 0.0069	13.0425	5.30	0.79	16.676
12	IMP	0.416027 ± 0.0263	41.6027	6.32	2.52	19.189
13	z	1.677398 ± 0.0442	167.7398	2.64	10.19	16.319
14	z	0.012632 ± 0.0010	1.2632	7.77	0.08	21.967
Total		16.456440 ± 0.0265	1645.6440	0.16	100.0	
5 MOP + 8 MOP + IsoP		10.252239 ± 0.0875	1025.2239	0.85	62.3	

* percentage of total coumarins detected in the extract considered as 100%, ^&^-RSD (%) = (SD/mean) × 100; UM: umbelliferon; 8 MOP: xanthotoxin; IsoP: isopimpinellin; 5 MOP: bergapten; IMP: imperatorin; u: calculated/umbelliferon; x: calculated/xanthotoxin; y: calculated/bergapten; and z: calculated/imperatorin.

**Table 4 molecules-29-02874-t004:** Quantitative analysis of main methoxyfuranocoumarins in the PLE/ASE extracts with the use of different solvents and increasing temperature (SPE/HPLC/DAD assay).

Temp. (°C) Solvent	mg/100 g Dry Weight of Plant Substance (mean ± SD) */(% = g/100 g DW) ^&^		
	8 MOP	IsoP	5 MOP
dichloromethane			
Dex/50	187.91 ± 1.75/0.19	303.94 ± 2.11/0.30	129.20 ± 2.34/0.13
Dex/70	205.28 ± 2.06/0.21	311.04 ± 3.01/0.31	141.15 ± 1.53/0.14
Dex/90	227.36 ± 1.23/0.23	327.73 ± 3.70/0.33	156.33 ± 2.78/0.16
Dex/110	238.79 ± 3.02/0.24	333.53 ± 1.65/0.33	168.19 ± 1.54/0.17
Dex/130	259.53 ± 3.41/0.26	346.53 ± 2.76/0.35	178.45 ± 3.02/0.18
methanol			
Mex/50	296.05 ± 1.24/0.30	358.76 ± 2.01/0.36	203.56 ± 1.40/0.20
Mex/70	330.16 ± 1.41/0.33	374.56 ± 2.09/0.38	227.01 ± 2.54/0.23
Mex/90	351.37 ± 2.34/0.35	397.25 ± 1.98/0.40	241.60 ± 2.03/0.24
Mex/110	354.25 ± 2.60/0.35	403.59 ± 2.62/0.40	243.58 ± 1.78/0.24
Mex/130	368.03 ± 4.51/0.37	404.14 ± 3.54/0.40	253.05 ± 3.55/0.25

* *n* = 6, ^&^ (%) in DW of the plant material; Dex: dichloromethane; Mex: methanol extracts.

**Table 5 molecules-29-02874-t005:** Calculated selectivity index (SI) of isopimpinellin (SI is the ratio of the mean IC50 in a normal cell culture to the mean IC50 in a cancer cell culture).

Normal Cells / Tumor Cells (IC_50_)	HSF / Saos-2	HSF / HOS	HSF / RPMI8226	HSF / U266	HSF / SW620	HSF / HT-29
SI *	9.62	1.27	3.91	4.88	0.57	4.88

* SI > 1.0 indicates a drug’s efficacy against cancer cells is greater than its toxicity against normal cells.

## Data Availability

All of the data are available on reasonable request from the corresponding author.

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
