# Peer review of "Quantitative Analysis of Isopimpinellin from Ammi majus L. Fruits and Evaluation of Its Biological Effect on Selected Human Tumor Cells"

_molecules, 2024, doi:10.3390/molecules29122874_

Round 1

Reviewer 1 Report

Comments and Suggestions for Authors

This paper described extraction and quantitative analysis of several coumarins from Ammi majus L.  fruits and the effects of isopimpinellin (The most abundant of them) on the cell viability in various cell lines.

The paper is generally well-written, although the discussion is too extensive and should be shortened to align with the content of the work. On the other hand, the isolation and quantitative analysis of coumarins using ASE/HPLC/DAD are widely described techniques that do not offer any novelty. Similarly, many furanocoumarins have demonstrated cytotoxicity in various cell lines.

Regarding the results, it is not possible to differentiate between the cytostatic and cytotoxic effects of isopimpinellin by studying only its effect on cell viability using MTT. The results described in the paper are confusing. Selectivity is used to justify an antiproliferative effect, and this is not correct. Furthermore, the selectivity studies have not been conducted properly because while the tumor cells were treated with isopimpinellin for 96 hours, the HSF cells were treated for 48 hours.

For all these reasons, the paper should not be published in its current form and additional studies are necessary.

1)       To determine whether the activity of isopimpinellin is cytotoxic or cytostatic, other studies such as flow cytometry, caspase activity determination, or cell DNA content analysis need to be conducted.

2)       To study the effects of isopimpinellin on cell viability in both tumor and non-tumor (HSF) cells over a similar period of time, so that the results can be comparable to determine selectivity.

Reviewer 2 Report

Comments and Suggestions for Authors

In this work entitled, “Quantitative analysis of Isopimpinellin from Ammi majus L. fruits and evaluation of its biological effect on selected human tumour cells” Bartnik et al., presented a quantitative HPLC approach for evaluation of IsoP isolated from the dichloromethane extract of the fruits prepared by ASE approach. The authors also assessed the in vitro cytotoxic effect against a number of cancer cells using MTT assay. Overall, the work is not only interesting but could be valuable to readers. Authors should be commended on the design, organization and clear presentation of the work in a manner that would be accessible to most readers. This study can add to the growing body of knowledge on how to approach the preparation/isolation of bioactive ingredients in a manner that is both efficacious and eco-friendly. Assessment of the biological activity is also notable, with sufficient number of different cancer cells. To further improve the quality of this work, authors are encouraged to consider the comments presented below.

Abstract

-Please articulate the aim of the work in one sentence at the beginning of the Abstract

-‘CPC’ should be written in full.

-Key findings, especially pertaining to the anti-proliferative effect of Iso-P should be supported with some data in the Abstract.

-Lines 57 – 64: Specify which part of the plant was used in the reports being reviewed in this section.

Section 2.1.

-What informed the selection of chloroform, methanol and dichloromethane as solvents of choice, despite their non-GRAS (generally regarded as safe) status?

-What is responsible for the better extraction efficiency of methanol vis-à-vis chloroform and dichloromethane besides the obvious difference in polarity.

-Given that ethanol is even a bit more polar than methanol, a GRAS solvent, and inexpensive why was ethanol not explored as a potential ASE solvent in this work?

-Lines 113 – 114: Authors should clarify that ‘extraction efficiency’ was in terms of extraction yield.

-Lines 114-128: Some would argue that comparing ASE with Soxhlet extraction or maceration is a bit unfair. A proper comparison should be with similar technique aimed to improve extraction efficiency such as ultrasound-assisted, microwave-assisted or supercritical fluid extraction. In other words, how does the present ASE approach compare with these enhanced extraction techniques?

-How was the RSD for extraction calculated?

-Authors can delete Figure 3. It would be more meaningful instead for Figure 2 to be modified. The Figure should feature the HPLC chromatogram of the extract (A) overlaid with HPLC chromatogram of the reference compounds (B)

-Lines 175 – 179: If your point of ASE was to improve ‘extraction efficiency’ it would make sense to use the methanol extract rather than the dichloromethane extract for isolation of IsoP. Besides, the argument that the methanol extract contains ballast does not negate this for three reasons. 1. Every extract contains ballast, and the authors did not provide a contrary evidence why dichloromethane extract would be better in this regard. 2. With higher content of IsoP in the methanol extract, its should have been the logical choice if greater recovery of the targeted compound is desired. 3. While the methanol extract may contain additional polar compounds as impurities, the dichloromethane extract presumably contains non-polar compounds as impurities as well. Authors should provide a compelling response for selecting dichloromethane extract for isolation of IsoP beside avoiding some of the ‘polar impurities’ in the methanol extract.

-Lines 227 – 238: There are well-established guidelines for assessing the cytotoxic activity of plant extracts as well as purified compounds from plants. In other words, there is a specific range of IC50 values which can be considered cytotoxic, moderately cytotoxic, and non-cytotoxic. Based on the presentation here (Lines 227 – 238) it appears the authors did not use any specific guideline. If the contrary is the case, authors should please provide the guideline used in evaluating the antiproliferative effect of IsoP and revise this section accordingly.

-Lines 236 – 238  “The other cell lines with IC50 between 84 -110 μM demonstrated a moderate tolerance to inhibitory action of isopimpinellin.” The meaning of this statement is not clear. Please re-phrase it for improved clarity.

-Lines 246 – 247 “A favourable SI > 1.0 indicates a drug with efficacy against cancer cells 246 greater than toxicity against normal cells [35].” It is questionable that a cytotoxic drug with favorable selectivity would have an SI value of >1.0. This yardstick is certainly low and should be verified.

-Line 505: The formula presented here seem to depict yield of the extract rather than the extraction efficiency.

-Line 585: What was the LC equipment, accessories and parameters used together with MS in the compound identification?

-The Conclusions section should be improved. Authors should highlight the principal findings, stress why these are important by providing specific implications. Also, authors should mention some of the challenges/limitations encountered in the work. As an extraction/preparative approach for IsoP, how feasible is the prospect to scale-up the preparation of this compound using the approach presented in this work? At best, cytotoxic efficacy of IsoP form this study can be categorized as moderate (IC50 value = 40.05 µM against Saos-2). What is the prospect of this agent and how can it be improved for anticancer treatment?

-Authors should consider including computational or in silico studies a convenient option to explore the potential anticancer mechanism of IsoP.

-The writing in this work is extremely poor. It is absolutely important for this manuscript to be thoroughly edited for English language. A professional English language editor with background in medicinal/natural product chemistry would be preferable.

Thanks.

Comments on the Quality of English Language

Extensive editing of English language is mandatory for this work.

Round 2

Reviewer 1 Report

Comments and Suggestions for Authors

The authors have conducted the additional experiments required to study the cytostatic/cytotoxic effect of isopimpinellin, and the quality of the work has been improved.

However,  this revised version contains numerous typographical errors: line 245 "acive" instead of "active," line 289 "saos-2" instead of "Saos-2," line 333 R3 instead of R4, line 348 "Isopimpinellin" instead of "isopimpinellin" and many others. Therefore, a detailed review must be conducted.

Reviewer 2 Report

Comments and Suggestions for Authors

The authors have made substantial changes to the manuscript and their response was mostly satisfactory. Nonetheless, some areas require further attention.

The response provided with respect to selecting the dichlorodiphenyltrichloroethane extract as opposed to the methanol extract is not convincing and should be properly addressed.

Secondly, a compound with selectivity of 1 is clearly not selective. It simple means that the compound is as toxic to healthy cells as it is toxic to cancer cells. This aspect of the work should be revised appropriately with clear understanding of the meaning of drug selectivity.

Last but not the least, authors are encouraged to seek for assistance with respect to improving the English language quality of the writeup.

Thanks.

Comments on the Quality of English Language

Meticulous editing of English language is required.
